# Molecular Characterization and Functional Analysis of Odorant-Binding Proteins in *Ectropis grisescens*

**DOI:** 10.3390/ijms26104568

**Published:** 2025-05-10

**Authors:** Fangmei Zhang, Haohan Sun, Shubao Geng, Shibao Guo, Zhou Zhou, Hongzhong Shi, Xuguo Zhou, Xiangrui Li

**Affiliations:** 1College of Agronomy, Xinyang Agriculture and Forestry University, Xinyang 464000, China; zhangfm@xyafu.edu.cn (F.Z.); 2024180001@xyafu.edu.cn (H.S.); 2018180002@xyafu.edu.cn (S.G.); 2008180026@xyafu.edu.cn (S.G.); 2021180005@xyafu.edu.cn (Z.Z.); 1988180008@xyafu.edu.cn (H.S.); 2Department of Entomology, School of Integrative Biology, College of Liberal Arts & Sciences, University of Illinois Urbana-Champaign, Urbana, IL 61801, USA; xgzhou@illinois.edu; 3State Key Laboratory for Biology of Plant Diseases and Insect Pests, Institute of Plant Protection, Chinese Academy of Agricultural Sciences, Beijing 100193, China

**Keywords:** *Ectropis grisescens*, odorant-binding protein, qRT-PCR, binding ability

## Abstract

Insect odorant-binding proteins (OBPs) are promising molecular targets for developing novel pest management strategies by modulating chemoreception-driven behaviors. The tea gray geometrid *Ectropis grisescens* (Lepidoptera, Geometridae) is a major pest in tea plantations, causing substantial economic losses in China. In this study, we identified 18 OBPs from *E. grisescens* antennal transcriptome. All of the encoded proteins possessed N-terminal signal peptides and conserved cysteine residues, behaviors which are characteristic of insect OBPs. Phylogenetic analysis categorized these proteins into plus-C, minus-C, and classic OBP subfamilies. MEME motif analysis identified conserved sequence features potentially involved in odor detection. Tissue- and sex-specific expression profiling showed that *EgriGOBP1-2*, *OBP3*, *OBP8*, and *OBP13* were highly expressed in the antennae of both sexes, suggesting roles in olfactory communication. Among them, *EgriGOBP1-2*, *OBP3*, and *OBP13* exhibited similar expression levels between males and females, while other *EgriOBPs* were predominantly expressed in the legs, wings, or other tissues, indicating additional physiological functions beyond chemoreception. To investigate functional specificity, we selected antenna-enriched EgriGOBP2 for ligand-binding analysis. Fluorescence binding assays demonstrated that EgriGOBP2 exhibited broad binding affinity toward 8 of 12 host volatiles and 11 of 12 plant essential oil-derived volatiles. These combined findings lay the foundation for mechanistic studies of chemical recognition in *E. grisescens* and provide insights into the development of ecologically friendly pest control alternatives.

## 1. Introduction

Olfaction plays a crucial role in various insect behaviors, including host location, mate selection, and oviposition site identification. These behaviors rely on a highly sophisticated and sensitive olfactory system that enables insects to perceive chemical signals from the environment [1]. Several chemosensory gene families, including odorant-binding proteins (OBPs), chemosensory proteins (CSPs), odorant receptors (ORs), gustatory receptors (GRs), ionotropic receptors (IRs), and sensory neuron membrane proteins (SNMPs), have been demonstrated to be involved in the chemoreceptive process [2]. Among these, OBPs are believed to play a fundamental role in the initial steps of odorant recognition. They are primarily responsible for binding, solubilizing, and transporting odor molecules across the aqueous sensilla lymph to specific ORs on the dendrite membrane of olfactory neurons, thereby initiating chemical signal transduction and triggering olfactory behavior in many insects [3,4].

OBPs are small, water-soluble globular proteins with molecular weights ranging from 10 to 30 kDa. They are highly concentrated in the hydrophilic lymph of insect olfactory sensilla and are characterized by six highly conserved cysteine residues forming three disulfide bridges [5]. Based on distinct conserved cysteine patterns, insect OBPs can be classified into four subfamilies: classic OBPs (6 conserved cysteines), minus-C OBPs (4 conserved cysteines), plus-C OBPs (8 conserved cysteines), and atypical OBPs (9–10 conserved cysteines) [3]. In Lepidoptera, OBPs are also typically categorized into two subfamilies: pheromone-binding proteins (PBPs), which primarily bind pheromone components [6], and general odorant-binding proteins (GOBPs), which primarily bind plant volatiles [7]. However, numerous studies have demonstrated that PBPs can also recognize host plant volatiles [8], while GOBPs exhibit a strong affinity for sex pheromones [9]. Both PBPs and GOBPs belong to the classic OBPs group, sharing the defining structural characteristics of classic OBPs, including six conserved cysteine residues.

Since the first OBP was identified in the wild silk moth *Antheraea polyphemus* [10], a large number of OBPs have been discovered through genome and transcriptome sequencing across diverse insect orders, including Diptera [11], Hymenoptera [12], Lepidoptera [13], and Hemiptera [14]. Previous studies have demonstrated that most insect OBPs are highly expressed in the antennae based on qPCR analyses, indicating their crucial role in chemoreception. Moreover, OBPs have been found to be highly expressed in other tissues beyond their role in chemodetection, such as wings [15], legs [16], and reproductive organs [13]. Additionally, more robust evidence including OBP-odorant binding affinity studies [17], gene knockdown experiments [18], and in vivo trans-species tests [19] has revealed that OBPs perform different physiological roles in chemoreception, development, and insecticide resistance.

Tea is a perennial evergreen plant widely cultivated in China, India, Japan, Ceylon, Sri Lanka, Indonesia, and several African countries [20]. In China, tea production plays a significant role in the agricultural economy, with an industry output exceeding USD 2.08 billion, as reported by the FAO (http://www.fao.org, accessed on 9 August 2024) in 2022. However, tea plants are highly susceptible to herbivorous pests, which threaten the yield and quality. The tea gray geometrid *Ectropis grisescens* Warren (Lepidoptera: Geometridae) is one of the most serious defoliators of tea plants due to its wide distribution, voracious feeding behavior, and high reproductive capacity [21,22]. This pest significantly reduces tea plant growth, productivity, and quality, causing substantial economic losses in tea plantations [23]. Chemical pesticides are predominantly used to control *E. grisescens* outbreaks, but long-term overuse has led to significant resistance development in pest populations [24]. Consequently, there is an urgent need for novel, effective, and environmentally friendly strategies for the sustainable management of this pest.

Given the significant roles of OBPs in insect chemical communication, olfactory-based pest control strategies have been explored and successfully applied. For example, RNAi-mediated silencing of *BtabOBP3* significantly altered the host plant preference of *Bemisia tabaci* by affecting its response to β-ionone, a key plant volatile [25]. In *Grapholita molesta*, the knockdown of *GmolOBP7* significantly reduced electroantennogram (EAG) responses to the host volatile 1-dodecanol [26]. Similarly, depletion of the *OBP10* or *OBP22* gene in *Aedes aegypti* resulted in a significant reduction in female fecundity and fertility [27]. *E. grisescens* is known to be highly attracted to specific host plant volatiles and sex pheromones [28], making its olfactory system a potential target for behavior-based pest management strategies.

In the present study, 18 candidate EgriOBP genes were identified based on previous antennal transcriptome sequencing [29]. Sequence alignment, phylogenetic analysis, and motif pattern analysis were performed to characterize these molecules, and quantitative real-time PCR (qRT-PCR) was used to assess their expression in various tissues of both sexes. Additionally, EgriGOBP2, an antennae-enriched GOBP gene, was selected for further functional characterization. Its binding affinity to different host plant volatiles and plant essential oil-derived compounds was evaluated using fluorescence competitive binding assays. These findings provide insights into the molecular basis of olfaction in *E. grisescens* targets and may facilitate the development of novel, olfaction-based pest control strategies.

## 2. Results

### 2.1. Identification of OBP in E. grisescens

A total of 18 OBP genes (GenBank accession numbers: ON380510–ON380527) were identified in the antennal transcriptome of *E. grisescens*, comprising 2 GOBPs and 3 PBPs (Table 1). All of the OBP genes had intact open reading frames (ORFs) with lengths ranging from 417 to 999 base pairs (bp). Their complementary DNAs (cDNAs) encoded proteins consisting of 130–188 amino acids (aa), with molecular weights (Mw) ranging from 15.0 to 38.7 kilodaltons (kDa) and isoelectric points (pI) ranging from 4.4 to 8.6. The signal peptides of four EgriOBPs (EgriGOBP2, EgriOBP2, EgriOBP4, and EgriOBP11) were predicted at the N-terminal. The BLASTx results indicated that 13 EgriOBPs shared relatively high amino acid identities (>98%) with the identified genes of *Ectropis obliqua* (Table 1). The sequence similarity among EgriOBPs ranged from 0.58% to 62.05%. The EgriPBPs and EgriGOBPs exhibited an overall sequence identitiy of 52% (Appendix A).

Based on the number and pattern of conserved cysteine residues, the 18 EgriOBPs were classified into three subfamilies. EgriOBP1 and EgriOBP12 were classified as minus-C OBPs, lacking the C2 and C5 cysteine residues. EgriOBP2 and EgriOBP10 were categorized as plus-C OBPs, possessing three additional conserved cysteines—one between C4 and C5 and two downstream of C6—in addition to the six conserved cysteines. The remaining 14 EgriOBPs were classified as classic OBPs, all of which conformed to the profile of “C1-X31-33-C2-X4-C3-X43-C4-X21-C5-X8-C6” (where X represents any amino acid) (Figure 1).

### 2.2. Phylogenetic Analysis

The phylogenetic tree was constructed based on 95 OBP amino acid sequences (Figure 2). The results indicated that these EgriOBPs were well segregated from each other and clustered into distinct branches. The EgriGOBP and EgriPBP subfamilies were grouped into the same major clade but formed separate clusters. The identified EgriPBP1-3 and EgriGOBP1-2 were clustered into the PBP and GOBP families, respectively. EgriOBP1 and EgriOBP12 were clustered into the minus-C OBP family, while EgriOBP2 and EgriOBP10 were grouped into the plus-C OBP family. The remaining EgriOBPs were scattered across the phylogenetic tree, with each well segregated and clustered into different branches.

### 2.3. Motif Pattern Analysis

The 10 motifs of the EgriOBPs identified in *E. grisescens* were discovered using the MEME program (Figure 3A). Among these, motifs 1 and 2 were conserved the most, being present in 15 out of 18 OBPs, except for EgriOBP1-2 and EgriOBP10. Phylogenetic analysis revealed that EgriPBP1-3, comprising three potential PBPs, exhibited two distinct motif patterns. EgriPBP2 and EgriPBP3 shared the motif arrangement 8-3-1-5-4-2-7, whereas EgriPBP1 lacked motif 3 at its N-terminus compared with EgriPBP2-3. The two potential GOBPs, EgriGOBP1 and EgriGOBP2, had identical motif patterns, with each missing motif 7 at the C-terminus compared with EgriPBP2-3. Interestingly, the motif patterns of EgriOBP1 contained only motif 2, whereas EgriOBP12 had three motifs (10-1-2), resembling EgriOBP13. The two potential plus-C OBPs, EgriOBP2 and EgriOBP10, exhibited only motif 9. Four EgriOBPs (EgriOBP5, 6, 7, and 9) exhibited the same motif order (1-6-2). EgriOBP3-4, 8, and 11 each showed only motifs 1 and 2 and were positioned identically (motif 1 at the N-terminal and motif 2 at the C-terminal) (Figure 3A).

### 2.4. Expression Profiles of EgriOBPs

The abundance of EgriOBPs in the antenna transcriptomes of female and male *E. grisescens*, as normalized reads in terms of FPKM, is presented in Figure 3B. Among all EgriOBPs, EgriOBP8 exhibited the highest abundance (FPKM = 78,085.30 in male antennae and 24,411.06 in female antennae), followed by EgriPBP2 (FPKM = 24,300.93 in males and 6398.80 in females), EgriPBP3 (FPKM = 15,941.07 in males and 4597.19 in females), and EgriGOBP2 (FPKM = 4780.95 in males and 10,897.12 in females).

The tissue- and sex-specific expression profiles confirmed that most EgriOBPs were detected in both the antennae and non-olfactory tissues (Figure 4). Five OBPs (*EgriGOBP1-2*, *OBP3*, *OBP8*, and *OBP13*) were highly enriched in the antennae, with significantly higher expression compared with other tissues (*p* < 0.05). Notably, *EgriOBP8* exhibited an expression level 2.38 times higher in male antennae than in female antennae (*p* < 0.01). In contrast, *EgriGOBP1-2*, *OBP3*, and *OBP13* showed similar expression levels between the sexes, with no significant differences (*p* > 0.05). Four OBPs (*EgriPBP1-2*, *OBP2*, *OBP4,* and *OBP5*) were predominantly expressed in the male antennae compared with other tissues (*p* < 0.05), while *EgriPBP3*, *EgriOBP1*, and *EgriOBP6* exhibited low expression levels in both sexes.

Furthermore, the expression of certain OBPs was highly elevated in non-olfactory tissues. Specifically, *EgriPBP1* and *EgriOBP5-7* showed pronounced female-biased expression in the head, while *EgriPBP2* and *EgriOBP11-12* were predominantly expressed in the wings of females. Two OBPs, *EgriOBP1* and *EgriOBP12*, displayed significantly higher expression levels in male abdomens, with *EgriOBP1* showing a 6.57-fold increase and *EgriOBP12* showing an 8.96-fold increase compared with females (*p* < 0.01). Additionally, *EgriOBP9* exhibited high expression levels in the legs, with significant sex differences (*p* < 0.05).

### 2.5. Expression and Purification of EgriGOBP

To investigate the functional role of OBPs in olfaction, EgriGOBP2 was selected for expression and characterization. The recombinant EgriGOBP2 was successfully expressed in *Escherichia coli* BL21 (DE3) cells following induction with 1 mM IPTG for 10 h. SDS-PAGE analysis confirmed the presence of the expected EgriGOBP2 band (Figure 5A). The purified recombinant EgriGOBP2 protein had an approximately molecular weight of 20 kDa, with a final yield of 0.26 mg/mL of soluble protein. After the removal of the His-tag, the molecular weight of the EgriGOBP2 protein was determined to be 15.24 kDa. The purified protein was subsequently used for fluorescence-based ligand-binding assays.

### 2.6. Fluorescence Competitive Binding Analyses of EgriGOBP2

The binding affinity of EgriGOBP2 for the fluorescent probe 1-NPN was assessed using binding curves and Scatchard plot analysis, revealing strong binding between EgriGOBP2 and 1-NPN with a dissociation constant (Kd) of 9.28 µM. The analyses of the binding curves and Scatchard plots indicated that 1-NPN was a suitable probe for subsequent competitive binding analyses (Figure 5B). To evaluate the ligand-binding specificity of EgriGOBP2, fluorescence displacement assays were conducted with 24 candidate ligands, comprising 12 host volatiles and 12 plant essential oil-derived volatiles. The competitive fluorescence binding curves indicated that most tested ligands reduced the relative fluorescence intensity of the EgriGOBP2/1-NPN complex, suggesting significant binding interactions (Figure 6A,B). The median inhibitory concentration (IC_50_) and dissociation constant (*K*_i_) values were derived from the binding curves, and they are summarized in Table 2 and Figure 6.

Among the 12 host volatiles, EgriGOBP2 exhibited strong binding affinity for eight compounds (*K*_i_ < 40 µM). The compounds nerolidol, citral, and linalool exhibited the highest binding affinities to EgriGOBP2, with *K*_i_ values of 7.93, 8.23, and 10.76 µM, respectively. Benzyl acetate, methyl salicylate, and n-hexyl alcohol also exhibited considerable binding affinity, with *K*_i_ values below 22 µM. Benzyl alcohol exhibited a moderate binding affinity (*K*_i_ =27.05 µM), whereas 1-penten-3-ol showed a weak binding affinity (*K*_i_ > 40 µM). Notably, four host volatiles, E-2-hexenal, benzaldehyde, decanal, and nonanal, did not significantly reduce the fluorescence intensity, indicating negligible binding to EgriGOBP2 (Figure 6A; Table 2).

Interestingly, 11 of the 12 tested plant essential oil-derived volatiles exhibited high binding affinities (*K*_i_ < 22 µM) for EgriGOBP2. Among these, (S)-(+)-carvone demonstrated the strongest binding, with a *K*_i_ value of 4.84 µM, followed by 4-allylanisole, eugenol, and eucalyptol, with *K*_i_ values of 7.31, 8.38 and 8.61 µM, respectively (Figure 6B; Table 2). Additionally, cis-3-hexenyl acetate, cis-3-hexenyl caproate, ocimene, and cis-3-hexenyl tyrate exhibited moderate binding, with *K*_i_ values of 13.07, 13.99, 17.98, and 18.62 µM, respectively. Carvacrol (*K*_i_ = 21.29 μM) and geranyl acetate (*K*_i_ = 22.31 μM) also showed moderate binding affinities. However, EgriGOBP2 did not bind to cinnamaldehyde, as evidenced by its inability to displace 1-NPN fluorescence.

## 3. Discussion

In this study, 18 candidate OBP genes were identified from the antennal transcriptome of *E. grisescens*. The number of OBP genes varies significantly among insect species, reflecting differences in their ecological adaptations and olfactory requirements. For instance, 38 OBPs were reported in *Spodoptera litura* [30], 10 in *Aphis glycines* [31], 54 in *Athetis dissimilis* [13], only 8 in *B. tabaci* MED [32]. The number of OBPs identified in *E. grisescens* is comparable to those found in the antennal transcriptomes of *Manduca sexta* (18) [33] and *Pieris rapae* (14) [34]. Such variations among different species may be attributed to evolutionary pressures, including adaptation to diverse chemosensory environments, specialization in host plant detection, or increased ligand-binding sensitivity, which can lead to gene duplication, functional diversification, and gene loss.

The sequence similarity among OBPs within the same species is generally low, likely due to rapid evolutionary divergence in response to environmental pressures [3]. Consistent with this, the amino acid sequence identity among EgriOBPs ranged from 0.58% to 62.05%, suggesting substantial functional diversification driven by ecological adaptations [35]. However, within the Lepidopteran-specific subgroups, namely GOBPs and PBPs, sequence similarity exceeded 50%, indicating their functional conservation across evolutionary timescales.

Based on the number of conserved cysteine residues, the 18 EgriOBPs were classified into three subfamilies: 14 classic OBPs, 2 plus-C OBPs, and 2 minus-C OBPs. This distribution closely resembles that of *A. glycines* (7 classic OBPs, 1 plus-C OBPs, and 1 minus-C OBPs) [31] and *Cydia pomonella* (30 classic OBPs, 4 plus-C OBPs, and 4 minus-C OBPs) [36]. Alterations in the spacing of cysteine residues may alter the three-dimensional structure of an OBP, affecting its binding properties and enabling it to interact with a distinct array of odor molecules or pheromones. This implies that EgriOBPs might have evolved specialized functions in the olfactory system, potentially dedicated to detecting specific information chemicals that are crucial for the survival and reproduction of *E. grisescens*. In the present study, the predominance of classic OBPs (78%) indicates that these proteins play a crucial role in detecting host plant volatiles and pheromones. Furthermore, Gene Ontology (GO) annotation and Kyoto Encyclopedia of Genes and Genomes (KEGG) pathway analyses of the antennal transcriptome identified multiple olfactory-related functions, such as localization, signaling and responses to stimuli in the biological process ontology, as well as signal transduction and environmental adaptation in KEGG [29]. These findings further support the hypothesis that the identified EgriOBPs participate in diverse chemosensory processes in *E. grisescens*.

The multiple sequence alignment results showed that the EgriOBPs were divided into three families: classic OBPs, plus-C OBPs, and minus-C OBPs. Consistent with the findings from the neighbor-joining tree, the identified EgriOBPs clustered into several distinct branches, aligning with previous studies [37]. Notably, the EgriGOBP and EgriPBP subfamilies were assorted into the same branch but formed separate clusters, suggesting that they diverged from a common ancestral gene and subsequently diverged due to speciation and reproductive isolation. This finding aligns with the well-established functional divergence between these two subfamilies; GOBPs are primarily involved in recognizing plant volatiles, whereas PBPs are specialized for detecting sex pheromone components. Furthermore, EgriGOBPs and EgriPBPs exhibited high bootstrap values with their homologs in *E. obliqua* (EoblGOBPs and EoblPBPs, respectively), indicating strong evolutionary conservation between these sibling species and suggesting potential similarities in their chemoreceptive functions.

The motif patterns play a crucial role in regulating OBP functions, affecting their ability to bind semiochemicals [32]. MEME motif analysis revealed that EgriOBPs exhibited diverse motif patterns. Among them, EgriPBP1-3 displayed two distinct motif arrangements, with EgriPBP1 lacking motif 3. This structural variation suggests potential functional divergence among EgriPBPs. Similar differences in PBP motif patterns have also been reported in other insects, such as *P. rapae* [33]. Additionally, a study on *S. litura* demonstrated that its three PBP genes exhibited varying binding affinities toward female sex pheromones [38]. Moreover, EgriGOBPs had different motifs compositions compared with EgriPBPs, further supporting the functional differentiation between these two OBP subfamilies. Interestingly, motif 7 was exclusively present in EgriPBP1-3, suggesting that this conserved motif may play a unique role in sex pheromone binding in insect olfaction, warranting further investigation.

Tissue-and sex-specific expression profiles revealed variations in EgriOBP expression across different tissues and between sexes. Most EgriOBPs were detected in both antennae and non-olfactory tissues, consistent with previous reports [39,40]. Notably, five EgriOBPs (*EgriGOBP1-2* and *OBP3*, *8*, and *13*) were significantly expressed in the antenna, indicating their potential involvement in chemical communications in *E. grisescens*. These results were consistent with the expression patterns observed in other Lepidoptera species, such as *P. rapae* [34] and *Spodoptera exempta* [27]. Additionally, three EgriPBPs were expressed at significantly higher levels in males than in females, consistent with the widely accepted notion that PBPs are male-biased and function in pheromone detection [41]. Similar male-biased expression patterns have been reported in the PBPs of *S. litura* [30] and *Plutella xyllotella* [42]. Meanwhile, the FPKM value analysis showed that *EgriPBP2* was the most abundantly expressed of the three EgriPBPs in male antennae, suggesting that it may play a predominant role in sex pheromone detection. Further studies are necessary to elucidate the precise function of EgriPBP2 in the olfactory system of *E. grisescens*.

Interestingly, *EgriOBP1* and *12* exhibited higher expression in the abdomens, which might reflect other possible physiological roles, such as the perception of mating partners and oviposition sites for *E. grisescens*. Additionally, *EgriOBP9* was predominately expressed in the female legs, a pattern also observed in *Adelphocoris lineolatus* [43] and *Apis cerana* [44]. It had been reported that insect legs sense chemical signals when the insects land on a host. For example, the legs of Drosophila can detect food resources and non-volatile pheromones [45]. Thus, it is reasonable to hypothesize that *EgriOBP9* may facilitate host location and reproduction in *E. grisescens*. Aside from that, *EgriPBP2* and *EgriOBP11-12* exhibited female-biased expression in the wings, consistent with the findings for *HarmOBP3* and *HarmOBP6*, which are expressed in the wings of *H. armigera*, indicating a positive correlation between expression levels and flight capacity [46]. However, whether OBPs contribute to flight performance, such as wing development, muscle physiology, or oxidative stress regulation, remains speculative. Therefore, further investigation is required to explore the potential roles of EgriOBPs beyond chemoreception, such as using RNA interference (RNAi) or CRISPR-mediated knockdown of specific OBPs in wing tissue combined with flight performance assays. This approach would help assess whether these proteins play an active role in flight-related physiology.

Moreover, although the majority of the EgriOBPs identified in this study belong to the classic OBPs subfamily, several candidates exhibited distinctive transcript enrichment patterns or motif features that may hint at specialized functional roles. For example, five EgriOBPs (*EgriGOBP1-2* and *OBP3*, *8*, and *13*), which exhibited antenna-biased expression patterns, may play a crucial role in the initial detection of odor molecules, being involved in binding and transporting these molecules to olfactory receptors and thereby facilitating the transduction of olfactory signals. In addition to transcript enrichment phenomena, EgriOBPs possess unique functional domain features that may confer special functions. For example, EgriPBP1-3 displayed distinct motif arrangements, which could alter EgriPBP’s binding affinity or specificity for particular odor molecules. This enables them to recognize and bind specific chemical signals such as mating or host location. Different OBPs may have evolved unique functional adaptations to meet the diversity of insect olfactory perception. By concentrating on these OBPs, we can propose more precise functional hypotheses to direct future research.

Numerous previous studies have shown that GOBPs, with highly conserved structures, generally play a vital role in the perception of plant volatiles in Lepidoptera [47,48]. In the present study, we selected antennal-enriched EgriGBOP2 to confirm the binding abilities pf host volatiles and some plant essential oil-derived volatiles using fluorescence-based competitive binding assays. According to the binding test results, we found that EgriGOBP2 displayed a broad spectrum of binding to 8 out of 12 host volatiles and 11 out of 12 plant essential oil-derived volatiles with different structural characteristics, which aligns with previous reports on OBP38 in *Riptortus pedestris* [7] and OBP9 in *Spodoptera litura* [49].

In our study, EgriGOBP2 was found to bind citral, nerolidol, linalool, benzyl acetate, and methyl salicylate from host-derived odorants with quite high affinities (*K*_i_ < 20 μM), similar to what has been observed in other GOBP homologies. For example, GOBP2 in *Ectropis obliqua*, a sibling geometrid of *E. grisescens*, has been shown to competitively bind to host odors such as benzaldehyde and methyl salicylate [50]. Likewise, SfruGOBP2 from *Spodoptera frugiperda* exhibited a broader ligand-binding spectrum, binding 21 volatiles and four insecticides [51]. Similarly, ScinGOBP2 in *Semiothisa cinerearia* exhibited strong binding abilities for 8 out of 27 host plant volatiles [48]. Additionally, the recombinant PsauGOBP2 from the variegated cutworm *Peridroma saucia* demonstrated binding affinities (6 µM ≤ *K*_i_ ≤ 13 µM) for the host plant volatiles phenylethyl acetate, β-myrcene, and dodecanol [52]. Furthermore, previous studies have demonstrated that GOBPs are expressed in sensilla basiconica [50], which are primarily responsible for sensing plant odors. In contrast, certain candidate odors in our study (e.g., E-2-hexenal, benzaldehyde and nonanal) did not exhibit binding affinity to EgriGOBP2. This suggests that these compounds may serve as ligands for other OBPs present in *E. grisescens*. As a result, we propose that EgriGOBP2 likely plays a role in the olfactory system by binding and transporting plant volatiles, as evidenced by its strong binding affinity for host volatiles.

Plant essential oils have exhibited contact, fumigant, and repellent toxicity against adults or larvae in various insect species and have been widely used as bioactive agents. For example, eucalyptol, the second most abundant volatile from *Rosmarinus officinialis* L, showed significant repellent activity against three thrip species (*Frankliniella occidentalis*, *Frankliniella intonsa*, and *Thrips palmi* Karny) in Y-tube olfactometer bioassays [53]. Carvone, a natural insect repellent, has been found to be strongly repellent to some insects, such as *Sitophilus zeamais* [54]. Eugenol is one of the active compounds that frequently appears in the essential oils of plants with repellent activity [55,56]. In the present study, 11 of 12 plant essential oil-derived volatiles showed strong bounding to EgriGOBP2, as determined by ligand-binding assays. Among these, (S)-(+)-carvone, eugenol, eucalyptol, and 4-allylanisole (*K*_i_ < 10 μM) demonstrated strong binding affinities to EgriGOBP2.

OBPs have been considered promising molecular targets for screening odorous compounds with attractant or repellent properties, as they exhibit high binding capacities for behaviorally active compounds in various insect species. Furthermore, in previous studies, OBPs exhibited high binding capabilities for behavioral attractants and repellents in certain insect species. For instance, CpalOBP2 from *Chrysopa pallens* exhibited strong and high binding affinities for both farnesene and its corresponding alcohol, farnesol, which had been confirmed to elicit significant and strong repellent behavioral responses using a glass Y-tube olfactometer in *C. pallens* [57]. Similarly, carvacrol was identified as a strong binder of AgamOBP5 in *Anopheles gambiae* [58]. In *Riptortus pedestris*, RpedOBP38 was found to bind plant essential oil-derived volatiles with repellent activity, including (+) -4-terpineol, (−)-carvone, and carvacrol [7], which have been reported to exhibit strong repellent properties against multiple insect species, such as *Halyomorpha halys* [59]. Notably, EgriGOBP2 exhibited a distinct binding affinity for all tested plant essential oil-derived volatiles, with *K*_i_ values below 20 µM, except for cinnamaldehyde and geranyl acetate. Given these results, future studies should evaluate the behavioral response of *E. grisescens* to compounds that exhibit high binding affinities for EgriGOBP2 to facilitate the identification of slow-release agents that attract or repel *E. grisescens* and to develop push-pull pest control strategies.

## 4. Materials and Methods

### 4.1. Insect Rearing, Sample Collection, and RNA Extraction

The larvae of *Ectropis grisescens* were collected from Mount Zhenlei (32.065° N, 114.145° E) in Xinyang, Henan province, China and reared on fresh tea leaves at 24 ± 1 °C under a relative humidity of 60% ± 5% with an L:D = 16:8 photoperiod. Different tissue samples were dissected and collected, including antennae (50 pairs), heads (from 10 individuals without antennae), abdomens (from 5 individuals), legs (from 20 individuals), and wings (from 5 individuals) from male and female adults. All samples were frozen in liquid nitrogen immediately and stored at −80 °C.

TRIzol reagent (Life Technologies, Carlsbad, CA, USA) was used to extract the total RNA from the collected samples, following the manufacturer’s instructions, and cDNA synthesis was performed using TransScript One-step gDNA Removal and cDNA synthesis SuperMix (TianGen, Beijing, China). The resultant cDNA samples were stored at −20 °C.

### 4.2. Identificantion and Analysis of EgriOBPs

The putative EgriOBPs genes were identified by searching a previously published antennal transcriptome database of *E. grisescens* (BioProject number: PRJNA784387, available in the NCBI SRA database) [29]. Homology-based searches were conducted using BLASTX (https://blast.ncbi.nlm.nih.gov/Blast.cgi, accessed on 14 September 2024) against the NCBI non-redundant (nr) protein database with an E value cut-off of 10^−5^. To ensure the accuracy of candidate identification, both sequence similarity and the domain structure were considered. Specifically, all candidate sequences were further examined for the presence of the conserved odorant-binding protein (OBP) domain using the Pfam database (http://pfam.xfam.org, accessed on 18 September 2024). Only sequences containing the characteristic OBP domain were retained, and the presence of conserved cysteine residues was noted. All retrieved OBP sequences were manually curated to remove incomplete entries and redundant entries. The open reading frames (ORFs) were identified with the ORF Finder Tool at the NCBI website (https://www.ncbi.nlm.nih.gov/orffinder/, accessed on 23 September 2024), and putative signal peptides were predicted with the SignalP 5.0 Server (http://www.cbs.dtu.dk/services/SignalP, accessed on 5 October 2024). The theoretical isoelectric point (pI) and molecular weight (Mw) of the predicted EgriOBPs were calculated using the Compute pI/Mw tool on the ExPASy server (https://web.expasy.org/compute_pi/, accessed on 5 October 2024). Multiple sequence alignments were performed using Clustal integrated into Jalview (v2.11.20) with the default parameters. Phylogenetic analysis was constructed using the neighbor-joining method implemented in MEGA 11.0 software with 1000 bootstrap replications, and the resulting trees were visualized using the Evolview online platform (https://www.evolgenius.info/, accessed on 18 December 2024). Motif pattern analysis was performed using the online program MEME (http://meme-suite.org/tools/meme, accessed on 25 December 2024) [60], with the parameters set to a minimum motif width of 6, maximum motif width of 10, and maximum number of 10 motifs to find.

### 4.3. Quantitative Real-Time PCR (qRT-PCR) Assay and Data Analysis

The expression levels of putative EgriOBPs in different tissues were evaluated using qRT-PCR. Gene-specific primers were designed using Primer Premier 5.0 software (Premier Biosoft International, Palo Alto, CA, USA; Appendix A) and synthesized by Sangon Biotech (Shanghai, China). Then, qRT-PCR was performed on an Applied Biosystems 7500 Fast Real-Time PCR System (Applied Biosystems, Carlsbad, CA, USA) using TB Green Premix Ex Taq (Tli RNaseH Plus) (TaKaRa, Beijing, China) in a 20 µL reaction volume according to the manufacturer’s instructions, with initial denaturation at 95 °C for 30 s, followed by 40 cycles of 95 °C for 5 s and 60 °C for 34 s, and a final step of 95 °C for 15 s, 60 °C for 1 min, and 95 °C for 15 s, with a final melting curve analysis step. Each reaction was performed in three biological replicates and three technical replications. Glyceraldehyde-3-phosphate dehydrogenase (GAPDH) was selected as the reference gene for normalization.

The relative expression levels of putative EgriOBPs were calculated using the comparative 2^−ΔΔCT^ method, and we compared the significance of each candidate EgriOBP across various tissues using a one-way analysis of variance (ANOVA, *p* < 0.05). Differences in OBP expression between female and male adults in different tissues (*p* < 0.05) were assessed using a two sample *t*-test in SAS statistical software 9.2 (SAS Institute Inc., Cary, NC, USA).

### 4.4. Expression and Purification of Recombinant EgriOBPs

Specific primers were designed with restriction enzyme sites BamHI and EcoRI using Primer Premier 5.0 (Appendix A). The target sequence was excised from a pMD^TM^ 19-T vector (TaKaRa) with the specific endonuclease and inserted into a digested pET30a (+) (LMAI Bio., Shanghai, China) vector. The recombinant plasmids were transferred into *E. coli* BL21 (DE3) competent cells (TransGen Biotech., China) for target protein expression. The recombinant protein expression and purification processes were similar to those described by Zhai et al. [61]. The recombinant proteins were purified with Ni-NTA His-Bind Resin (7Sea, Shanghai, China), eluted with 10 mL elution buffer (20 mM Tris-HCl, pH7.4, and 250 mM NaCl) with gradient concentrations (50, 100, 200, and 250 mM) for washing, and desalted using a dialysis membrane. The purity and size of the proteins were confirmed via SDS-PAGE, and the concentrations of the proteins were measured with the protocols of a BCA protein assay kit (Cwbio Biotech, Beijing, China).

### 4.5. Fluorescence Competitive Binding Assay

The binding affinities of target proteins were measured with an F-7000 fluorescence spectrophotometer (Hitachi, Tokyo, Japan) using a 1 cm light path quartz cuvette, and 1-N-phenyl-naphthylamine (1-NPN) was used as a fluorescent probe. The excitation wavelength was 337 nm, and the emission spectrum ranged from 370 to 550 nm. A total of 24 volatile compounds, including host volatiles [62] and plant essential oil-derived volatiles [7,59,63], were selected for fluorescence competitive binding assays (Table 2). First, the binding affinities of 1-NPN to the target proteins were determined. The protein solutions (2 µM in 20 mM Tris-HCl, pH = 7.4) were titrated with 1-NPN to final concentrations ranging from 0 to 16 µM. The binding curves were linearized using Scatchard plots, based on fluorescence intensity values relative to the maximum fluorescence emissions. Next, each ligand (at concentrations of 0–16 µM, prepared at a 1 mM stock concentration) was added to the 2 µM EgriGOBP2/1-NPN mixture, and competitive binding curves were plotted based on the fluorescence intensity values from three replicates.

The dissociation constants (*K*_i_) of each ligand were calculated with the following equation: *K*_i_ = [IC_50_]/(1 + [1-NPN]/K_1-NPN_). IC_50_ is the ligand concentration replacing 50% of the initial fluorescence intensity of the EgriGOBP2/1-NPN mixture, [1-NPN] represents the free concentration of 1-NPN, and *K*_1-NPN_ is the dissociation constant of the EgriGOBP2/1–NPN complex, which was calculated using Scatchard plots of the binding data in GraphPad Prism 8.0 (GraphPad Software, San Diego, CA, United States). The binding affinity of ligands to EgriGOBP2 was considered extremely strong (*K*_i_ ≤ 6 µM), strong (6 µM < *K*_i_ ≤ 22 µM), moderate (22 µM < *K*_i_ ≤ 40 µM), or weak (*K*_i_ > 40 µM) [15].

## 5. Conclusions

In this study, we identified 18 EgriOBPs based on the previous antennal transcriptome database of *E. grisescens* and observed both tissue- and sex-specific expression patterns, indicating that these OBP genes may be involved in various aspects of chemical communications in *E. grisescens*. Furthermore, EgriGBOP2, which was highly expressed in the antennae of adult *E. grisescens*, was selected for functional characterization. The binding abilities of EgriGOBP2 for host volatiles and plant essential oil-derived volatiles were investigated using fluorescence competitive binding analyses. Our results demonstrated that EgriGOBP2 displayed a broad spectrum of binding affinities, binding 8 of 12 host volatiles and 11 of 12 plant essential oil-derived volatiles, suggesting its potential role in perceiving environmental chemical cues in *E. grisescens*. Our findings not only provide insight into the olfactory sensitivity of *E. grisescens* to host volatiles but also facilitate the identification of behavioral attractants and inhibitors derived from plant essential oils to support the development of ecologically friendly pest control strategies. Future studies should focus on elucidating the precise role of interactions between EgriGOBP2 and various compounds using a combination of behavioral assays, electrophysiological experiments, and RNA interference approaches.

## Figures and Tables

**Figure 1 ijms-26-04568-f001:**
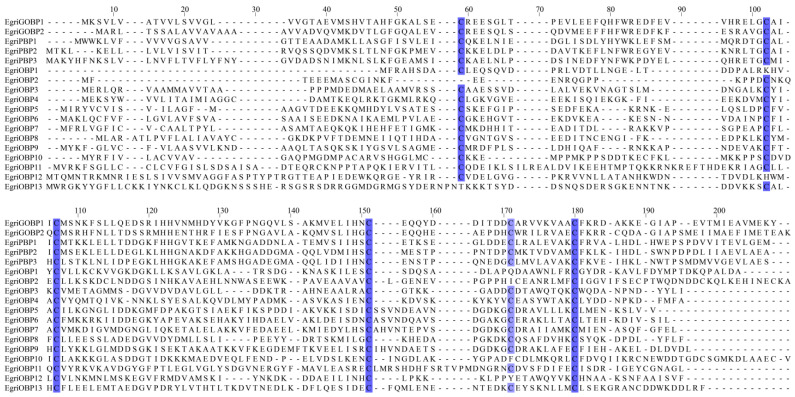
Multiple sequence alignments of *E. grisescens* OBPs. The six conserved cysteine residues are highlighted in blue.

**Figure 2 ijms-26-04568-f002:**
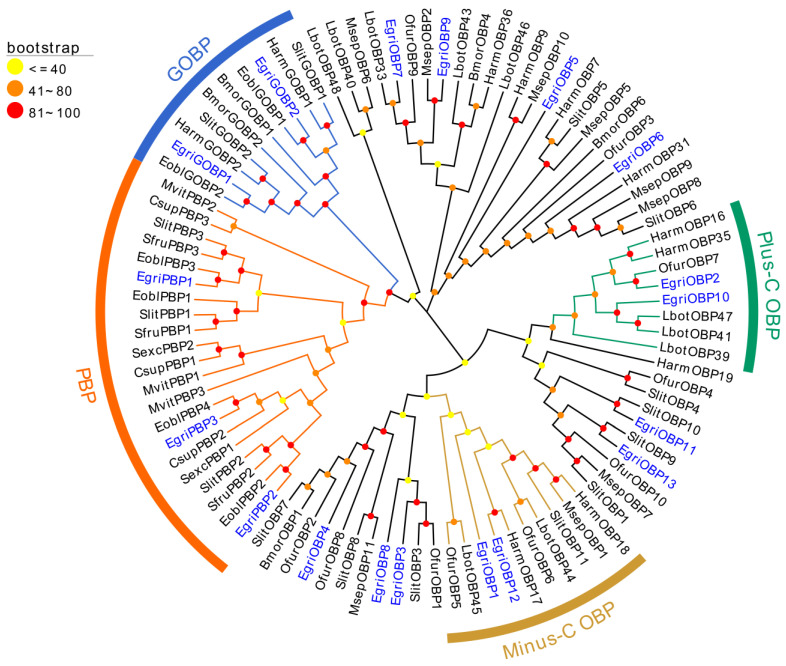
Phylogenetic tree of candidate EgriOBPs along with known Lepidoptera OBP sequence. Eobl = *Ectropis obliqua*; Harm = *Helicoverpa armigera*; Bmor = *Bombyx mori*; Mvit = *Maruca vitrat*; Csup = *Chilo suppressalis*; Slit = *Spodoptera litura*; Sfru = *Spodoptera frugiperda*; Sexc = *Scirpophaga excerptalis*; Ofur = *Ostrinia furnacalis*; Lbot = *Lobesia botrana*; Msep = *Mythimna separata*. The GenBank accession numbers and sequences of the 95 OBP proteins used in this phylogenetic analysis are listed in Appendix A.

**Figure 3 ijms-26-04568-f003:**
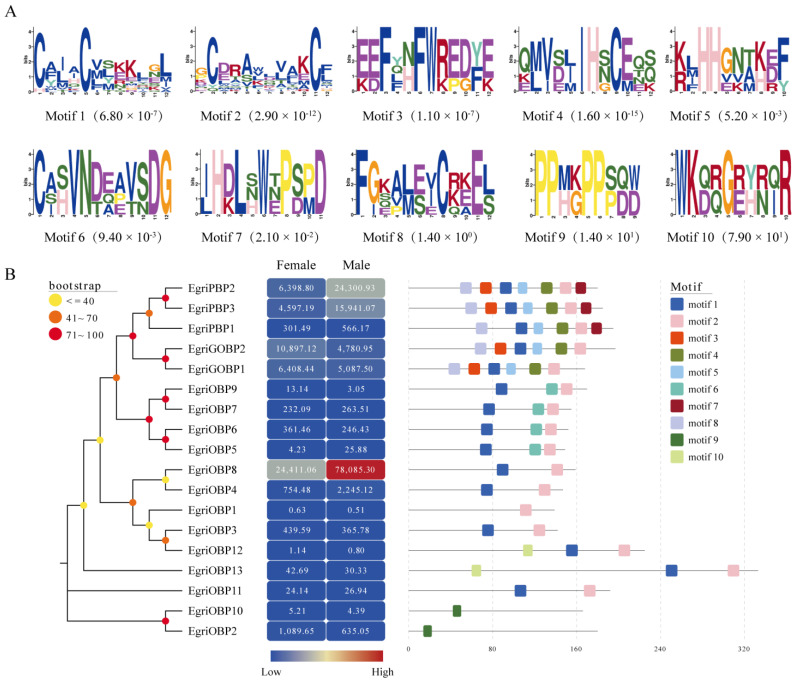
Phylogenetic analysis, expression levels, and motif patterns of EgriOBPs. (**A**) Ten conserved motifs identified by MEME, along with their respective E values. (**B**) The neighbor-joining tree of 18 EgriOBPs, with corresponding expression levels and motif patterns. Heat map illustrates the abundance of EgriOBPs in female and male *E. grisescens* antennal transcriptomes, represented in terms of fragments per kilobase of the exon model per million mapped reads (FPKM). Each column represents one sample, and each row represents one OBP gene. The intensity of the color reflects read abundance, with red indicating higher expression and blue indicating lower expression.

**Figure 4 ijms-26-04568-f004:**
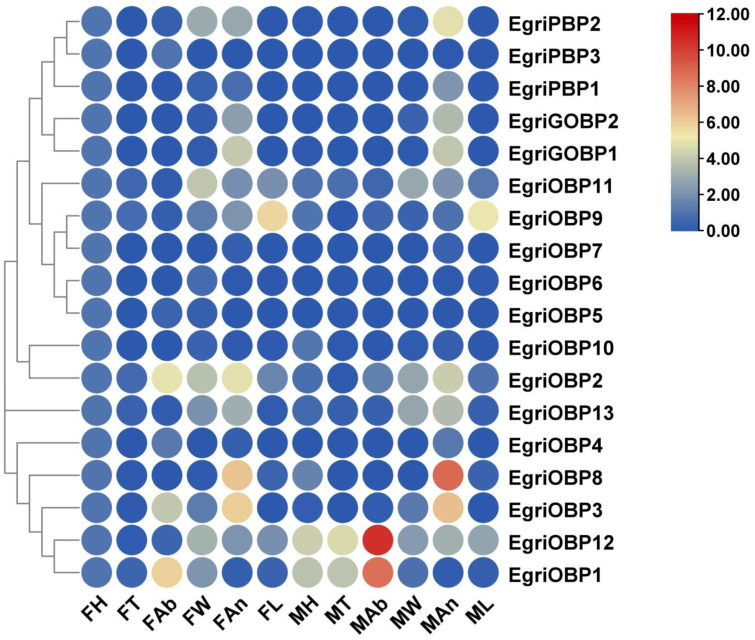
Tissue- and sex-specific expression profiles of EgriOBPs. Gene expression levels were normalized relative to female head expression (set to onefold). FH = female head; FT = female thorax; FAb = female abdomen; FW = female wing; FAn = female antennae; FL = female leg; MH = male head; MT = male thorax; MAb = male abdomen; MW = male wing; MAn = male antennae; ML = male leg.

**Figure 5 ijms-26-04568-f005:**
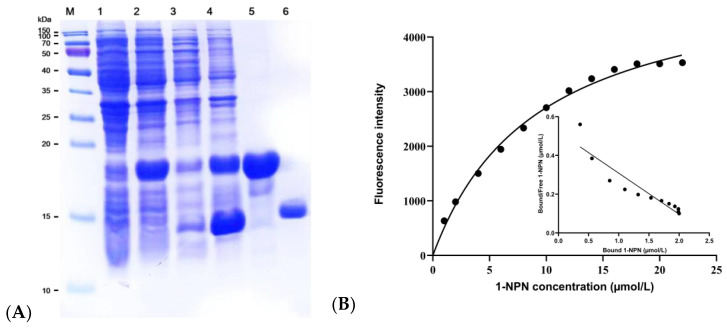
(**A**) SDS-PAGE analysis of recombinant EgriGOBP2 protein. M = protein molecular weight marker; Lane 1 = non-inducted pET32a (+)/EgriGOBP2 protein; Lane 2 = IPTG-induced pET32a (+)/EgriGOBP2 protein; Lane 3 = supernatant of pET32a (+)/EgriGOBP2 protein; Lane 4 = pellet of pET32a (+)/EgriGOBP2 protein; Lane 5 = purified pET32a (+)/EgriGOBP2 protein; Lane 6 = EgriGOBP2 protein after His-tag removal. (**B**) Binding curves of 1-NPN and corresponding linear Scatchard plot.

**Figure 6 ijms-26-04568-f006:**
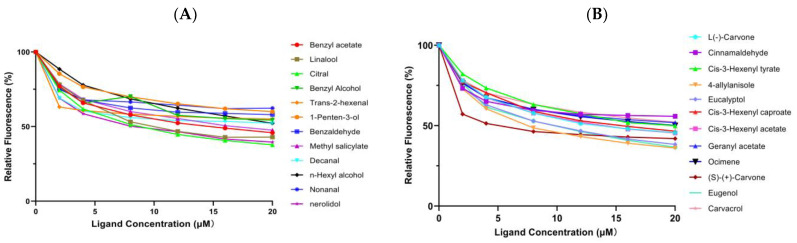
Binding assay results of 24 volatile compounds with recombinant EgriGOBP2. (**A**) Host volatiles. (**B**) Plant essential oil-derived volatiles. The *K*_i_ values of EgriGOBP2 for different tested compounds are listed in Table 2.

**Table 1 ijms-26-04568-t001:** Identification and bioinformatics analysis of odorant-binding protein (OBP) genes in *E. grisescens* antennal transcriptomes.

Gene Name	Accession Number	Length (bp)	ORF (aa)	Blastx Best Hit (Name and Species)	Accession Number	E Value	Identity (%)	Signal Peptide	pI	Mw (kDa)
EgriGOBP1	ON380526	504	160	general odorant-binding protein 2 [*Ectropis obliqua*]	ACN29681.1	2 × 10^−93^	100.00	NO	5.8	19.0
EgriGOBP2	ON380510	591	214	general odorant-binding protein 1 [*Ectropis obliqua*]	ACN29680.1	4 × 10^−92^	98.58	YES	5.9	22.2
EgriPBP1	ON380520	585	162	pheromone-binding protein 3 [*Ectropis obliqua*]	ALS03849.1	1 × 10^−91^	100.00	NO	4.9	21.7
EgriPBP2	ON380512	540	163	pheromone-binding protein 2 [*Ectropis obliqua*]	ALS03848.1	2 × 10^−89^	100.00	NO	5.3	19.9
EgriPBP3	ON380514	555	170	pheromone-binding protein 4 [*Ectropis obliqua*]	ALS03850.1	9 × 10^−119^	100.00	NO	6.1	20.8
EgriOBP1	ON380521	417	120	odorant-binding protein [*Semiothisa cinerearia*]	QRF70927.1	2 × 10^−72^	79.41	NO	8.3	15.0
EgriOBP2	ON380524	540	134	odorant-binding protein 10 [*Ectropis obliqua*]	ALS03858.1	5 × 10^−106^	100.00	YES	5.1	20.0
EgriOBP3	ON380515	426	138	odorant-binding protein [*Semiothisa cinerearia*]	QRF70921.1	3 × 10^−43^	81.58	NO	4.4	15.1
EgriOBP4	ON380517	441	142	odorant-binding protein 11 [*Ectropis obliqua*]	ALS03859.1	5 × 10^−92^	99.29	YES	8.6	16.8
EgriOBP5	ON380527	447	145	odorant-binding protein 18 [*Ectropis obliqua*]	ALS03866.1	3 × 10^−90^	98.62	NO	5.5	16.2
EgriOBP6	ON380523	456	150	odorant-binding protein 6 [*Ectropis obliqua*]	ALS03854.1	8 × 10^−92^	98.65	NO	5.3	16.4
EgriOBP7	ON380525	465	151	odorant-binding protein 14 [*Ectropis obliqua*]	ALS03862.1	2 × 10^−105^	100.00	NO	5.3	17.0
EgriOBP8	ON380513	477	156	odorant-binding protein 9 [*Ectropis obliqua*]	ALS03857.1	3 × 10^−96^	99.29	NO	4.4	17.5
EgriOBP9	ON380518	510	160	odorant-binding protein 17 [*Ectropis obliqua*]	ALS03865.1	7 × 10^−93^	100.00	NO	7.0	18.9
EgriOBP10	ON380511	498	163	odorant-binding protein OBP47 [*Lobesia botrana*]	AXF48744.1	6 × 10^−32^	52.17	NO	5.1	18.3
EgriOBP11	ON380516	576	184	odorant-binding protein 4 [*Ectropis obliqua*]	ALS03852.1	2 × 10^−125^	100.00	YES	6.5	21.5
EgriOBP12	ON380519	675	210	odorant-binding protein 18 [*Dendrolimus punctatus*]	ARO70177.1	7 × 10^−20^	40.50	NO	8.4	25.9
EgriOBP13	ON380522	999	331	odorant-binding protein [*Semiothisa cinerearia*]	QRF70922.1	1 × 10^−104^	75.74	NO	5.5	38.7

**Table 2 ijms-26-04568-t002:** Binding affinity of selected volatile compounds to recombinant EgriOBP2.

Ligands		Formula	CAS No#	Purity (%)	EgriGOBP2	
					IC_50_ (µM)	*K*_i_ (µM)
Host volatiles	Benzyl acetate	C_9_H_10_O_2_	140-11-4	99.00	14.35	12.95
	Linalool	C_10_H_18_O	78-70-6	97.00	11.92	10.76
	Citral	C_10_H_16_O	5392-40-5	95.00	9.12	8.23
	Benzyl alcohol	C_7_H_8_O	100-51-6	99.00	29.97	27.05
	E-2-hexenal	C_10_H_16_O	6728-26-3	98.00	-	-
	1-penten-3-ol	C_5_H_10_O	616-25-1	99.00	44.84	40.57
	Benzaldehyde	C_7_H_6_O	100-52-7	99.50	-	-
	Methyl salicylate	C_8_H_8_O_3_	119-36-8	98.00	17.27	15.59
	Decanal	C_10_H_20_O	112-31-2	98.00	-	-
	n-hexyl alcohol	C_6_H_14_O	111-27-3	98.00	23.48	21.20
	Nonanal	C_9_H_18_O	124-19-6	95.00	-	-
	Nerolidol	C_15_H_26_O	7212-44-4	98.00	8.79	7.93
Plant essential oil-derived volatiles	Cinnamaldehyde	C_9_H_8_O	104-55-2	95.00	-	-
	Cis-3-hexenyl tyrate	C_10_H_18_O_2_	16491-36-4	95.00	20.63	18.62
	4-allylanisole	C_10_H_12_O	140-67-0	98.00	8.10	7.31
	Eucalyptol	C_10_H_18_O	406-67-7	95.00	9.36	8.61
	Cis-3-hexenyl caproate	C_12_H_22_O_2_	31501-11-8	99.00	15.50	13.99
	Cis-3-hexenyl acetate	C_8_H_14_O_2_	3681-71-8	98.00	14.48	13.07
	Geranyl acetate	C_12_H_20_O_2_	105-87-3	97.00	24.72	22.31
	Ocimene	C_10_H_16_	13877-91-3	90.00	19.92	17.98
	(S)-(+)-carvone	C_10_H_14_O	2244-16-8	98.00	5.36	4.84
	Eugenol	C_10_H_12_O_2_	97-53-0	99.00	9.28	8.38
	Carvacrol	C_10_H_14_O	499-75-2	98.00	23.58	21.29
	L(-)-carvone	C_10_H_14_O	6485-40-1	99.00	12.87	11.62

“-” means that IC_50_ could not be calculated because no detectable *K*_i_ value was observed in the binding assay. The binding affinity of ligands to recombinant EgriGOBP2 was considered incredibly strong (*K*_i_ ≤ 6 µM), strong (6 µM < *K*_i_ ≤ 22 µM), moderate (22 µM < *K*_i_ ≤ 40 µM), or weak (*K*_i_ > 40 µM).

## Data Availability

The data that support the findings of this study are available from the corresponding author upon reasonable request.

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
