# Peer review of "Molecular Characterization and Functional Analysis of Odorant-Binding Proteins in Ectropis grisescens"

_ijms, 2025, doi:10.3390/ijms26104568_

Round 1
Reviewer 1 Report
Comments and Suggestions for Authors
In this manuscript, Zhang et al. identified 18 odorant-binding proteins (OBPs) from the antennal transcriptome of Ectropis grisescens. They further characterized the molecular features and binding affinities of the antenna-enriched EgriGOBP2. The findings provide a basis for future mechanistic studies on chemical recognition in E. grisescens. Overall, the methodology and results are clearly presented. However, a number of minor revisions are needed before the manuscript can be considered for publication.
Results
- Line 112:Revise “130 to 188 amino acids” to “130 to 188 amino acids (aa)”.
- Line 114:The sentence refers to “12 EgriOBPs,” but based on context, this number may actually be 13. Please verify this and check for consistency throughout the manuscript.
- Lines 121–122:Ensure consistent formatting of the OBP subfamily names: either “Minus-C/Plus-C” or “minus-c/plus-c.” Please unify the style throughout the manuscript.
- Line 115:The name obliqua appears for the first time and should be written in full as Ectropis obliqua. Please verify the correct and consistent naming of all insect species mentioned.
- Line 135:The text states that 95 OBP amino acid sequences were used, while Figure 2 refers to 93 OBP proteins. Please confirm whether the correct number is 93 or 95 and revise accordingly.
- Line 154:“12 out of 15” is mentioned, but earlier it was stated that there are 18 OBPs. Should this be “15 out of 18”? Please double-check the data.
- Line 239:“EgriOBP2” should be corrected to “EgriGOBP2.”
- Line 241:Please remove the duplicated phrase “Host volatiles.”
Discussion
- Line 327:Gene names such as EgriGOBP1-2, OBP3, OBP8, and OBP13 should be italicized following scientific naming conventions.
Figures
- Figure 3:Please standardize the use of “motif” (capitalize or lowercase consistently) across the manuscript.
Tables
- Table 2:There is an unnecessary hyphen that should be removed. Additionally, please revise the formatting of bold text in the table to ensure consistency and clarity.
Materials and Methods
- Lines 438 and 442:Please confirm whether “qRT-PCR” or “RT-qPCR” is the correct and consistent term to be used.
- Lines 477–478:The sentence “Candidate ligands were screened for attractant/repellent potential based on our plant essential oil library” seems out of context and lacks connection to the surrounding text. Please clarify or remove it.
Author Response
Reviewer #1:
In this manuscript, Zhang et al. identified 18 odorant-binding proteins (OBPs) from the antennal transcriptome of Ectropis grisescens. They further characterized the molecular features and binding affinities of the antenna-enriched EgriGOBP2. The findings provide a basis for future mechanistic studies on chemical recognition in E. grisescens. Overall, the methodology and results are clearly presented. However, a number of minor revisions are needed before the manuscript can be considered for publication.
RESPONSE: We appreciate reviewer’s comments.
Comments to Results
Comments 1: Line 112:Revise “130 to 188 amino acids” to “130 to 188 amino acids (aa)”.
Response 1: Revisions have been made following reviewer’s suggestions. (Line 113)
Comments 2: Line 114:The sentence refers to “12 EgriOBPs,” but based on context, this number may actually be 13. Please verify this and check for consistency throughout the manuscript.
Response 1: We have carefully checked the numbers and corrected them accordingly throughout the manuscript. (Line 116)
Comments 3: Lines 121–122:Ensure consistent formatting of the OBP subfamily names: either “Minus-C/Plus-C” or “minus-c/plus-c.” Please unify the style throughout the manuscript.
Response 3: We have unified the formatting of the OBP subfamily names throughout the manuscript.
Comments 4: Line 115:The name obliqua appears for the first time and should be written in full as Ectropis obliqua. Please verify the correct and consistent naming of all insect species mentioned.
Response 4: We have revised it to Ectropis obliqua and checked the consistency of all species names mentioned. (Line 117)
Comments 5: Line 135:The text states that 95 OBP amino acid sequences were used, while Figure 2 refers to 93 OBP proteins. Please confirm whether the correct number is 93 or 95 and revise accordingly.
Response 5: Thanks for your careful review. We have carefully reviewed the data and corrected the number in the revised manuscript.
Comments 6: Line 154:“12 out of 15” is mentioned, but earlier it was stated that there are 18 OBPs. Should this be “15 out of 18”? Please double-check the data.
Response 6: We have checked and corrected the numbers for consistency.
Comments 7: Line 239:“EgriOBP2” should be corrected to “EgriGOBP2.”
Response 7: We have revised “EgriOBP2” to “EgriGOBP2” in the manuscript.
Comments 8: Line 241:Please remove the duplicated phrase “Host volatiles.”
Response 8: We have deleted the redundant phrase.
Comments to Discussion
Comments 9: Line 327:Gene names such as EgriGOBP1-2, OBP3, OBP8, and OBP13 should be italicized following scientific naming conventions.
Response 9: We have revised the gene names to italicized formatting accordingly. (Lines 338-360)
Comments to Figures
Comments 10: Figure 3:Please standardize the use of “motif” (capitalize or lowercase consistently) across the manuscript.
Response 10: We have standardized “motif” to use lowercase consistently in the manuscript. (Line 171)
Comments to Tables
Comments 11:There is an unnecessary hyphen that should be removed. Additionally, please revise the formatting of bold text in the table to ensure consistency and clarity.
Response 11: We have corrected the hyphen issue and revised the formatting of bold text in Table 2.
Comments to Materials and Methods
Comments 12: Lines 438 and 442: Please confirm whether “qRT-PCR” or “RT-qPCR” is the correct and consistent term to be used.
Response 12: We have standardized the term to “qRT-PCR” throughout the manuscript.
Comments 13: Lines 477-478:The sentence “Candidate ligands were screened for attractant/repellent potential based on our plant essential oil library” seems out of context and lacks connection to the surrounding text. Please clarify or remove it.
Response 13: We have removed this redundant sentence, to improve clarity.
Reviewer 2 Report
Comments and Suggestions for Authors
Comments to the Authors:
Tracking no: ijms-3592162
Title: Molecular Characterization and Functional Analysis of Odor-ant-Binding Proteins in Ectropis grisescens
This research utilized transcriptomic data to identify and analyze 18 candidate OBP genes in the Ectropis grisescens. Through a combination of bioinformatic analyses and experimental validation, the functions of several OBP genes were further elucidated. These findings provide insights into the molecular basis of olfaction in E. grisescens targets and may facilitate the development of novel, olfaction-based pest control strategies. The manuscript is comprehensive, well-structured. However, I believe that this manuscript still needs to address the following issues before it can be considered for publication.
Comments:
Could the method used to identify OBP members be described in more detail? I noticed that there is no GFF file for Ectropis grisescens in NCBI, which suggests that the transcriptome would need to be assembled with Trinity.
In the Methods section, it only mentions using the antenna transcriptome to identify OBP genes, while the “Introduction” points out that OBPs have a broader range of functions. Does this mean that using only the antenna transcriptome to identify OBP genes might not be sufficiently rigorous?
Is it feasible to rely solely on BLAST? Since OBP genes contain a conserved domain, it is recommended to take both sequence similarity and the presence of this domain into account. Also, it is important to specify the parameters used in BLAST because, in my experience, the results can vary significantly depending on these parameters. In contrast, when searching for the domain, the number of candidate members tends to remain more consistent.
In the multiple sequence alignment section, please explain why these 18 OBP protein sequences do not start with the amino acid methionine (M). Typically, if a protein annotation is complete, it should begin with an M. Additionally, the manuscript mentions: “The sequence similarity among EgriOBPs ranged from 0.58% to 62.05%.” Why is the sequence similarity so low? Normally, sequences identified using BLAST exhibit similarity values above 20%.
The manuscript mentions RPKM twice but FPKM multiple times. Both are units for measuring expression levels, yet they differ. Please ensure consistency in their usage.
Author Response
Reviewer #2:
This research utilized transcriptomic data to identify and analyze 18 candidate OBP genes in the Ectropis grisescens. Through a combination of bioinformatic analyses and experimental validation, the functions of several OBP genes were further elucidated. These findings provide insights into the molecular basis of olfaction in E. grisescens targets and may facilitate the development of novel, olfaction-based pest control strategies. The manuscript is comprehensive, well-structured. However, I believe that this manuscript still needs to address the following issues before it can be considered for publication.
RESPONSE: We appreciate reviewer’s comments.
Comments 1: Could the method used to identify OBP members be described in more detail? I noticed that there is no GFF file for Ectropis grisescens in NCBI, which suggests that the transcriptome would need to be assembled with Trinity.
Response 1: We added a more detailed description of the identification method in Section 4.2. The transcriptome of Ectropis grisescens was assembled using Trinity with min_kmer_cov set to 2 and other parameters at default settings, as reported in our previous publication (Zhang et al., 2023). All CDS and protein sequences have been uploaded to NCBI database (Genbank ID: PRJNA784387). (Lines 454-465)
Zhang, F.M.; Chen, Y.J.; Zhao, X.C.; Guo, S.B.; Hong, F.; Zhi, Y.N.; Zhang, L.; Zhou, Z., Zhang. Y.H.; Zhou, X.G.; Li, X.R. Antennal Transcriptomic Analysis of Carboxylesterases and Glutathione S-transferases Associated with Odorant Degradation in the Tea Gray Geometrid, Ectropis grisescens (Lepidoptera, Geometridae). Front. Physiol. 2023, 14, 1183610.
Comments 2: In the Methods section, it only mentions using the antenna transcriptome to identify OBP genes, while the “Introduction” points out that OBPs have a broader range of functions. Does this mean that using only the antenna transcriptome to identify OBP genes might not be sufficiently rigorous?
Response 2: Thank you for your valuable comment. In this study, we specifically targeted antennal OBPs, as the antenna are the primary olfactory organs and antennal OBPs are typically the most abundant and functionally specialized for chemosensation. While OBPs may indeed have broader roles in gustation, hormone transport, and other physiological processes, our primary focus was to identify OBPs directly associated with olfactory function. Therefore, antennal transcriptomic data provide a reliable resource for this purpose. We acknowledge that using only antennal transcriptomes might result in the omission of OBPs predominantly expressed in non-olfactory tissues. However, since our study aims to understand the molecular basis of olfaction, this tissue-specific approach is appropriate and does not compromise our research objectives. Furthermore, we analyzed tissue- and sex-dependent expression patterns, and selected an antenna-enriched GOBP gene for further functional characterization. In future work, we plan to explore OBPs expressed in other tissues to complement our current findings.
Comments 3: Is it feasible to rely solely on BLAST? Since OBP genes contain a conserved domain, it is recommended to take both sequence similarity and the presence of this domain into account. Also, it is important to specify the parameters used in BLAST because, in my experience, the results can vary significantly depending on these parameters. In contrast, when searching for the domain, the number of candidate members tends to remain more consistent.
Response 3: We fully agree with the reviewer that relying solely on BLAST is insufficient for the accurate identification of OBP genes, and that the presence of the conserved OBP domain should also be taken into account. In the revised manuscript (Section 4.2), we have clarified our identification process accordingly. Specifically, we first performed a BLASTX search against the NCBI non-redundant (nr) protein database with an E-value threshold of 10-5, and subsequently examined all candidate sequences for the presence of the conserved odorant-binding protein (OBP) domain using the Pfam database. Only sequences containing the characteristic OBP domain were retained for downstream analyses. In addition, we have now specified the BLAST parameters used in our search to ensure reproducibility. We sincerely appreciate the reviewer’s insightful comments, which have helped us to improve the rigor and clarity of our methodology. (Lines 464-465)
Comments 4: In the multiple sequence alignment section, please explain why these 18 OBP protein sequences do not start with the amino acid methionine (M). Typically, if a protein annotation is complete, it should begin with an M.
Response 4: We carefully rechecked the sequences, removed redundant entries, and re-performed the multiple sequence alignment. The revised figure (Figure 1) has been updated in the manuscript.
Comments 5: Additionally, the manuscript mentions: “The sequence similarity among EgriOBPs ranged from 0.58% to 62.05%.” Why is the sequence similarity so low? Normally, sequences identified using BLAST exhibit similarity values above 20%.
Response 5: The reported sequence similarity range reflects pairwise comparisons among all identified EgriOBPs, including both closely related paralogs and more divergent family members. The low minimum value (0.58%) corresponds to comparisons between evolutionarily distant OBPs that retain only the conserved structural features (such as the six cysteine residues), but have diverged significantly in their primary sequences. This broad range in similarity highlights the inherent diversity within the OBP family. We also discuss the biological implications of this sequence diversity in the Discussion section.
Additionally, BLASTX analysis revealed that these OBPs shared 40.50%~100% amino acid identities with their orthologs in other Lepidopteran species (Table 1). Phylogenetic analysis further supported the classification of these sequences, as even the low-similarity OBPs clustered within well-established OBP clades.
Comments 6:The manuscript mentions RPKM twice but FPKM multiple times. Both are units for measuring expression levels, yet they differ. Please ensure consistency in their usage.
Response 6: We have corrected the inconsistency and standardized all expression level measurements to FPKM (Fragments Per Kilobase of transcript per Million mapped reads), which was used in our transcriptome analysis. (Lines 174, 345)
Reviewer 3 Report
Comments and Suggestions for Authors
This is an interesting and well-executed study that provides a solid molecular and functional characterization of odorant-binding proteins (OBPs) in an insect species of considerable economic importance. The work is clearly written, methodologically sound, and contributes valuable insights into the chemosensory mechanisms involved in host recognition and behavior. By focusing on a pest species associated with significant crop losses, the findings open promising avenues for the development of environmentally friendly, behavior-based pest control strategies. I commend the authors for a well-structured and scientifically rigorous contribution. I only suggest a few minor clarifications and improvements, which are outlined below.
Introduction
Lines 51-53: “They are highly concentrated in the hydrophilic lymph of insect olfactory sensilla, and are characterized by six highly conserved cytosine residues forming three disulfide bridges”
The term "cytosine" is incorrectly used here. As the authors know well, cytosine is a nitrogenous base and does not form disulfide bridges. The correct amino acid that forms disulfide bridges is cysteine, which contains a thiol (-SH) group. Therefore, the appropriate term in this context should be "cysteine residues", not "cytosine residues".
Lines 53-58:
“In Lepidoptera, OBPs are typically categorized into two sub families: pheromone-binding proteins (PBPs), which primarily bind pheromone com-ponents [6], and general odorant-binding proteins (GOBPs) for plant volatiles. Based on distinct conserved cysteine patterns, insect OBPs can be further classified into four subfamilies: Classic OBPs (6 conserved cysteines), Minus-C OBPs (4 conserved cysteines), Plus-C OBPs (8 conserved cysteines), and Atypical OBPs (9-10 conserved cysteines)”
It is mentioned that OBPs are functionally divided into PBPs and GOBPs, and later a structural classification based on conserved cysteine patterns is presented (Classic, Minus-C, Plus-C, and Atypical OBPs). While this dual classification—functional and structural—is highly relevant, the relationship between them is not explicitly addressed in the text.
In particular, pheromone-binding proteins (PBPs) are functionally distinct and well-studied in Lepidoptera, and are typically members of the Classic OBP subfamily. This connection is important and could help readers better understand how structural features underpin functional specialization in OBPs.
I suggest adding a brief clarification indicating that PBPs predominantly belong to the Classic OBP subfamily? This would resolve any potential confusion regarding whether PBPs are distributed across multiple structural OBP subtypes, and would strengthen the logical flow of the paragraph.
In the same paragraph:
“pheromone-binding proteins (PBPs), which primarily bind pheromone components, and general odorant-binding proteins (GOBPs) for plant volatiles”:
The sentence may inadvertently suggest that PBPs are not involved in binding volatile compounds. However, are not most Lepidopteran sex pheromones in fact volatile compounds? It would be more accurate to clarify that both PBPs and GOBPs bind volatile ligands, with their functional distinction relating to the nature of the ligand (pheromones vs. plant-derived odorants) rather than its volatility.
Discussion
Line 273-275:
“Such variations among different species may be attributed to evolutionary pressures, leading to gene duplication functional diversification, and gene loss”
The authors mention that species-specific variation may be attributed to evolutionary pressures leading to gene duplication and functional diversification. This is a valid point, but it would benefit from greater specificity. I suggest briefly elaborating on what kinds of selective pressures might drive these duplications—e.g., adaptation to diverse chemosensory environments, specialization in host-plant detection, or increased ligand-binding sensitivity. This would strengthen the argument and situate the observed genomic patterns in a clearer evolutionary context.
Line 287-291:
“The predominance of Classic OBPs (78%) in E. grisescens indicated that these proteins play a crucial role in detecting host plants volatiles and pheromones.Furthermore, Gene Ontology (GO) annotation and Kyoto Encyclopedia of Genes and Genomes (KEGG) pathway analyses of the antennal transcriptome identified multiple olfactory-related functions, further supporting the hypothesis that the identified EgriOBPs participate in diverse chemosensory processes in E. grisescens.”
This paragraph provides useful functional context for the predominance of Classic OBPs in E. grisescens. However, the discussion would be strengthened by clarifying whether the GO and KEGG annotations specifically support roles in pheromone binding, host plant detection, or other sensory modalities. Additionally, it may be worth noting whether any particular Classic OBPs show transcript enrichment or functional domain features that hint at specialized roles. This would help bridge the general annotation data with more specific functional hypotheses.
Line 340-344:
“Besides, EgriPBP2 and EgriOBP11-12 exhibited female-biased expression in the wings, which was consistent with the findings on HarmOBP3 and HarmOBP6, which are expressed in wings of H. armigera, indicating a positive correlation between expression levels and flight capacity [44]. Therefore, further investigation is warranted to explore the potential roles of EgriOBPs in non-olfactory tissues beyond chemoreception.”
The authors suggest that the female-biased expression of PBP2 and iOBP11-12 in wings may be functionally linked to flight capacity, drawing parallels with HarOBP3 and HarOBP6 in H. armigera. While this is a thought-provoking hypothesis, the causal relationship between OBP expression and flight performance remains speculative. I recommend clarifying whether these OBPs are known or predicted to have non-chemosensory roles, such as involvement in wing development, muscle physiology, or oxidative stress regulation. To strengthen the claim, the authors might consider proposing follow-up experiments, such as RNA interference (RNAi) or CRISPR-mediated knockdown of specific OBPs in wing tissue, combined with flight performance assays. This would help assess whether these proteins play an active role in flight-related physiology.
Line 373:
The sentence "Plant essential oil has displayed contact, fumigation, and repellent toxicity to adult or larvae in various insects and has been widely used as a bioactive agents" contains several grammatical issues. First, "plant essential oil" should be in the plural form ("plant essential oils") to reflect general usage. Second, "adult or larvae" should be corrected to "adults or larvae" for proper noun agreement. Finally, the phrase "a bioactive agents" mixes singular and plural forms; it should read simply "as bioactive agents." Consider revising the sentence as follows for clarity and grammatical accuracy.
Line 384-387:
"OBPs have been considered promising molecular targets for screening odorous compounds with attractant or repellent properties. Furthermore, previous studies have found that OBPs exhibit high binding capabilities for behavioral attractants and repellents previously in certain insect species.”
These two sentences state essentially the same idea—that OBPs can bind attractants and repellents. Consider merging for conciseness.
Line 389:
“elicited strong repellent behavioral responses in C. pallens”
This claim would benefit from a brief mention of experimental context (e.g., type of assay or behavioral setup) or citation of the method, if available.
Line 392:
Including instead inculding.
Author Response
Reviewer #3:
This is an interesting and well-executed study that provides a solid molecular and functional characterization of odorant-binding proteins (OBPs) in an insect species of considerable economic importance. The work is clearly written, methodologically sound, and contributes valuable insights into the chemosensory mechanisms involved in host recognition and behavior. By focusing on a pest species associated with significant crop losses, the findings open promising avenues for the development of environmentally friendly, behavior-based pest control strategies. I commend the authors for a well-structured and scientifically rigorous contribution. I only suggest a few minor clarifications and improvements, which are outlined below.
RESPONSE: We appreciate reviewer’s comments.
Comment to Introduction
Comment 1: Lines 51-53: “They are highly concentrated in the hydrophilic lymph of insect olfactory sensilla, and are characterized by six highly conserved cytosine residues forming three disulfide bridges”
The term "cytosine" is incorrectly used here. As the authors know well, cytosine is a nitrogenous base and does not form disulfide bridges. The correct amino acid that forms disulfide bridges is cysteine, which contains a thiol (-SH) group. Therefore, the appropriate term in this context should be "cysteine residues", not "cytosine residues".
Response 1: We have corrected the term "cytosine" to "cysteine" in the revised manuscript. (Line 51)
Comment 2: Lines 53-58:
“In Lepidoptera, OBPs are typically categorized into two sub families: pheromone-binding proteins (PBPs), which primarily bind pheromone com-ponents [6], and general odorant-binding proteins (GOBPs) for plant volatiles. Based on distinct conserved cysteine patterns, insect OBPs can be further classified into four subfamilies: Classic OBPs (6 conserved cysteines), Minus-C OBPs (4 conserved cysteines), Plus-C OBPs (8 conserved cysteines), and Atypical OBPs (9-10 conserved cysteines)”
It is mentioned that OBPs are functionally divided into PBPs and GOBPs, and later a structural classification based on conserved cysteine patterns is presented (Classic, Minus-C, Plus-C, and Atypical OBPs). While this dual classification—functional and structural-is highly relevant, the relationship between them is not explicitly addressed in the text.
Response 2: In the revised manuscript, we have clarified the relationship between PBPs/GOBPs and the structural classifications of OBPs, adding that PBPs are predominantly members of the Classic OBP subfamily. (Lines 52-62)
Comment 3: In particular, pheromone-binding proteins (PBPs) are functionally distinct and well-studied in Lepidoptera, and are typically members of the Classic OBP subfamily. This connection is important and could help readers better understand how structural features underpin functional specialization in OBPs.
I suggest adding a brief clarification indicating that PBPs predominantly belong to the Classic OBP subfamily? This would resolve any potential confusion regarding whether PBPs are distributed across multiple structural OBP subtypes, and would strengthen the logical flow of the paragraph.
Response 3: As recommended, we have added clarification that PBPs are primarily members of the Classic OBP subfamily, helping to avoid potential confusion regarding their structural classification. (Lines 52-62)
Comment 4: In the same paragraph:
“pheromone-binding proteins (PBPs), which primarily bind pheromone components, and general odorant-binding proteins (GOBPs) for plant volatiles”:
The sentence may inadvertently suggest that PBPs are not involved in binding volatile compounds. However, are not most Lepidopteran sex pheromones in fact volatile compounds? It would be more accurate to clarify that both PBPs and GOBPs bind volatile ligands, with their functional distinction relating to the nature of the ligand (pheromones vs. plant-derived odorants) rather than its volatility.
Response 4: We have revised the sentence to clarify that both PBPs and GOBPs can bind volatile ligands, with their functional distinction based on ligand nature (pheromones vs. plant-derived odorants), rather than volatility. Relevant references have also been added. (Lines 52-62)
Comment to Discussion
Comment 5: Line 273-275:
“Such variations among different species may be attributed to evolutionary pressures, leading to gene duplication functional diversification, and gene loss”
The authors mention that species-specific variation may be attributed to evolutionary pressures leading to gene duplication and functional diversification. This is a valid point, but it would benefit from greater specificity. I suggest briefly elaborating on what kinds of selective pressures might drive these duplications—e.g., adaptation to diverse chemosensory environments, specialization in host-plant detection, or increased ligand-binding sensitivity. This would strengthen the argument and situate the observed genomic patterns in a clearer evolutionary context.
Response 5: As suggested, we have elaborated on the types of selective pressures that might drive these variations, including adaptation to diverse chemosensory environments, host-plant detection, and increased ligand-binding sensitivity (Lines 280-282).
Comment 6: Line 287-291:
“The predominance of Classic OBPs (78%) in E. grisescens indicated that these proteins play a crucial role in detecting host plants volatiles and pheromones. Furthermore, Gene Ontology (GO) annotation and Kyoto Encyclopedia of Genes and Genomes (KEGG) pathway analyses of the antennal transcriptome identified multiple olfactory-related functions, further supporting the hypothesis that the identified EgriOBPs participate in diverse chemosensory processes in E. grisescens.”
This paragraph provides useful functional context for the predominance of Classic OBPs in E. grisescens. However, the discussion would be strengthened by clarifying whether the GO and KEGG annotations specifically support roles in pheromone binding, host plant detection, or other sensory modalities.
Response 6: We have clarified that the GO and KEGG analyses support roles in olfactory functions such as binding, signaling, and response to stimuli, with specific annotations linked to insect olfaction. We also included a reference to our previous study for further support (Lines 304-306).
Comment 7:Additionally, it may be worth noting whether any particular Classic OBPs show transcript enrichment or functional domain features that hint at specialized roles. This would help bridge the general annotation data with more specific functional hypotheses.
Response 7: Thanks for your careful review. As suggested, we have elaborate the specialized roles of several candidates which exhibit distinctive transcript enrichment patterns or motif features in discussion, in order to help bridge the general annotation data with more specific functional hypotheses. (Lines 295-300; Lines 368-381).
Comment 8:Line 340-344:
“Besides, EgriPBP2 and EgriOBP11-12 exhibited female-biased expression in the wings, which was consistent with the findings on HarmOBP3 and HarmOBP6, which are expressed in wings of H. armigera, indicating a positive correlation between expression levels and flight capacity [44]. Therefore, further investigation is warranted to explore the potential roles of EgriOBPs in non-olfactory tissues beyond chemoreception.”
The authors suggest that the female-biased expression of PBP2 and iOBP11-12 in wings may be functionally linked to flight capacity, drawing parallels with HarOBP3 and HarOBP6 in H. armigera. While this is a thought-provoking hypothesis, the causal relationship between OBP expression and flight performance remains speculative. I recommend clarifying whether these OBPs are known or predicted to have non-chemosensory roles, such as involvement in wing development, muscle physiology, or oxidative stress regulation. To strengthen the claim, the authors might consider proposing follow-up experiments, such as RNA interference (RNAi) or CRISPR-mediated knockdown of specific OBPs in wing tissue, combined with flight performance assays. This would help assess whether these proteins play an active role in flight-related physiology.
Response 8: We have rephrased this section to propose a hypothesis regarding the involvement of OBPs in flight performance, including potential roles in wing development, muscle physiology, and oxidative stress regulation. We also suggest future experiments using RNA interference (RNAi) or CRISPR-mediated knockdown to further investigate these roles (Lines 361-367).
Comment 9: Line 373:
The sentence "Plant essential oil has displayed contact, fumigation, and repellent toxicity to adult or larvae in various insects and has been widely used as a bioactive agents" contains several grammatical issues. First, "plant essential oil" should be in the plural form ("plant essential oils") to reflect general usage. Second, "adult or larvae" should be corrected to "adults or larvae" for proper noun agreement. Finally, the phrase "a bioactive agents" mixes singular and plural forms; it should read simply "as bioactive agents." Consider revising the sentence as follows for clarity and grammatical accuracy.
Response 9: Revisions have been made following reviewer’s suggestions (Lines 409-410).
Comment 10: Line 384-387:
"OBPs have been considered promising molecular targets for screening odorous compounds with attractant or repellent properties. Furthermore, previous studies have found that OBPs exhibit high binding capabilities for behavioral attractants and repellents previously in certain insect species.”
These two sentences state essentially the same idea—that OBPs can bind attractants and repellents. Consider merging for conciseness.
Response 10: We have merged these sentences for conciseness, now stating "OBPs have been considered promising molecular targets for screening odorous compounds with attractant or repellent properties, as they exhibit high binding capacities for behaviorally active compounds in various insect species." (Lines 421-423)
Comment 11: Line 389: “elicited strong repellent behavioral responses in C. pallens”
This claim would benefit from a brief mention of experimental context (e.g., type of assay or behavioral setup) or citation of the method, if available.
Response 11: Revisions have been made following reviewer’s suggestions (Lines 427-428).
Comment 12: Line 392: Including instead inculding.
Response 12: Revisions have been made following reviewer’s suggestions. (Line 431)
Round 2
Reviewer 2 Report
Comments and Suggestions for Authors
Thank you for your revisions and feedback. I have no more suggestions.